# Psychometric properties of The Baruth Protective Factors Inventory among nursing students

**Tayyebeh Ali-Abadi[1]◉, Siavash Talepasand[2]◉, Christopher Boyle[3]◉, Hamid Sharif Nia[4]***

1 Department of Nursing, Neyshabur Branch, Islamic Azad University, Neyshabur, Iran, 2 Department of Psychology and Educational Sciences, Semnan University, Semnan, Iran, 3 Education and Psychology, Graduate School of Education, University of Exeter, Exeter, United Kingdom, 4 School of Nursing and Midwifery Amol, Mazandaran University of Medical Sciences, Sari, Iran

◉ These authors contributed equally to this work.
* pegadis@yahoo.com

**Data Availability Statement:** All relevant data are within the manuscript and its Supporting Information files.

**Funding:** The author(s) received no specific funding for this work

## Abstract

The Baruth Protective Factors Inventory is an instrument which assesses the risk and protective factors behind resilience. However, there is no valid or reliable Persian instrument for measuring resilience amongst nursing students in Iran. This study aimed to evaluate the psychometric properties of the Persian version of The Baruth Protective Factors Inventory among nursing students. This methodological study was done in 2017. The participants were 200 nursing students who were randomly recruited from Neyshabur city. Construct validity was assessed using exploratory and confirmatory factor analysis. Reliability was also assessed through the internal consistency assessment method. Exploratory factor analysis indicated a three-factor structure for the Inventory which accounted for 48.21% of its total variance. Confirmatory factor analysis confirmed the first-order model. The internal consistency values of the Inventory and its domains were good, confirming its great internal consistency and reliability. The Persian version of the Baruth Protective Factors Inventory was found to have acceptable validity and reliability to assess resilience amongst nursing students. Therefore, the Persian version of The Baruth Protective Factors Inventory can help nursing authorities identify non-resilient students, promote their resilience, and thereby improve their academic and clinical performance. Nursing students' improved performance can positively affect both the quality of care and patient outcomes.

## Introduction

Nursing students are presented and need to cope with a wide range of challenges that are unique to their profession. For example, care delivery to patients with different physical and mental health problems, patients' death, and the risk of being infected by infectious diseases. These stressors cause nursing students different levels of anxiety, stress, and discomfort [1], also high levels of stress can affect them psychologically and cause sleep disorders, reduction in

**Competing interests:** The authors have declared that no competing interests exist.

their motivation for clinical activities, and undermine their self-confidence [2]. Simonelli-Muñoz et al (2018) found stress had worse academic performance in nursing students [3].

Resilience is an appropriate strategy for stress management [4]. Students' resilience can be promoted through providing themselves with protective factors such as a caring learning environment, learner-centered education, strong instructor-student relationship, and positive expectations Whereas teaching and learning processes focus merely on information delivery, students do not get psychologically prepared for clinical practice and thus, may be vulnerable to occupational stressors later in their professional lives [5].

However, resilience evaluation necessitates valid and reliable instruments, but due to diversity in resilience definitions, different instruments have been developed for its measurement [6]. These instruments include, but are not limited to, Brief Resilience Scale [7], Connor-Davidson Resilience Scale [8], Baruth Protective Factor Inventory [9], Resilience Scale for Adults [10], and the Brief Resilience Coping Scale [11]. However, as resilience is affected by many factors including context, time, age, and living conditions, its measurement tools also need to be context, time, and age-specific [12].

The conceptual framework of this study was based on the following definition of resilience: "an individualized process of development that occurs through the use of protective factors to successfully navigate perceived stress and adversities" [13]. In this framework, risk factors minimization and protective factors enhancement help promote the ability to cope with challenges in life. As Baruth quoted Wolff (1995) identifying protective stress-causing factors would play a significant role in clinical practices. Individuals with intervene risks, have further resilience while avoiding from environmental problems. When these people got further information about the reasons and protective strategies with the help of expert ones, more resilience and stability in their life would be predictable [9].

One of the research instruments that have been used to measure protective factors specifically in adults is the Baruth Protective Factors Inventory (BPFI). A significant strong point of BPFI theory is the rich theoretical basis for it. This valid and reliable inventory measures protective factors in an adult population through a review of resilience literature (resilience theories [9]. Assessing resilience through BPFI increases the chance of identifying concepts and variables which are important to nursing and health care practice. Therefore these variables will be clarified and important dimensions of the variables that should be measured will be determined [14]. Therefore, BPFI is particularly fascinating in nursing students to quickly identify stress which results in providing interventions to decrease the level of stress.

The BPFI discusses Protective Factors that are repeatedly used by nursing students [1],[15],[16], [5],[17],[18]. The BPFI has been used to examine resilience in various contexts, including among students at a public university in the southwest United States [19], the families of disabled children in Portugal [20], clinical and non-clinical groups [21]. There were no consistent findings regarding the factor structure of BPFI when tested using Confirmatory Factor Analysis (CFA) across different cultures [19, 20]. By considering these findings, previous validation studies in another setting, time and so on, are not warranted to remain valid in another time, culture or context. Therefore, regarding the deficiency of valid tools on nursing student setting the overall aim of this study was evaluation of the psychometric properties of the Persian version of BPFI among nursing students. Identifying the optimal inventory structure may encourage other researchers in this field to utilize this inventory, thus helping to develop the study of resilience in nursing student settings.

## Materials and methods

This methodological study was done in 2017 to evaluate the psychometric properties of the Persian BPFI. The study population consisted of all 376 bachelor's students who were studying

nursing in Azad and medical science Universities in Neyshabur, Iran, in the 2017–2018 academic years. A sample of 200 students was recruited via simple random sampling. For instruments with less than forty items, samples of 200 were reported to be adequate [22]. Data collection was performed using BPFI. Baruth et al. developed BPFI in 2002 based on the existing literature on resilience [9]. As BPFI has 16 items, they are scored on a five-point Likert-type scale from 1 ("Strongly disagree") to 5 ("Strongly agree"). Items 1 to 4 are reverse scored. The total BPFI score varied between 16 and 80, with higher scores indicating more protective factors [21].

## Procedure

After obtaining necessary permissions from its developers, the scale was translated from English into a Persian version based on the World Health Organization protocol of the forward-backward translation technique [23]. Two English-Persian translators were requested to independently translate the BPFI. An expert panel, consisting of some of this article's authors, two professional translators, and a Persian-English bilingual speaker assessed and unified the two translations and constructed a single Persian translation of inventory. Thereafter, a Persian-English translator was asked to back-translate the Persian BPFI into English. This English version of the Inventory was sent to Persian-English bilingual speaker, for confirmation of the correctness of translations and confirming the similarity of the achieved English BPFI with its original. Then, the primary Persian BPFI was piloted on a sample of 28 nursing students in small two to three-person groups. So, face validity assessed this way that Students were asked to comment on the clarity of the items. Their comments required us to revise two items (9 and 10 respectively). Finally, 200 nursing students completed the Persian BPFI and the collected data were used for construct validity and reliability assessment.

## Construct validity

**Exploratory factor analysis.**   The construct validity of the scale was evaluated using Maximum Likelihood Exploratory Factor Analysis (MLEFA) with promax rotation. The Kaiser–Meyer–Olkin (KMO) test and Bartlett's test of sphericity checked the appropriateness of the study sample and the factor analysis model. The test estimates sampling adequacy for each item in the model and the model as a whole. The indicator is a measure of the proportion of variance among variables that might be common variance [24]. KMO values between 0.8 and 1 indicate the sampling is adequate. In the first step, the latent factors were extracted based on Horn's Parallel Analysis[25]. The presence of a single item in the factor based on the $CV = 5.152 \div \sqrt{(n-2)}$ formula [26] was estimated to be approximately 0.4 (in the present formula, the CV is the number of extractable factors and n is the sample size of the study). According to the three indicator rule, at least three items must exist for each factor [27]. According to the three-indicator rule, there must be at least three items for each latent variable in the EFA [28]. Items with communalities less than 0.2 were excluded from the EFA [29].

**Confirmatory factor analysis.**   The extracted factors were evaluated by Confirmatory Factor Analysis (CFA) based on the Maximum Likelihood Method. CFA goodness of fit indexes includes the acceptable level of Parsimonious Normed Fit Index (PNFI), Parsimonious Comparative Fit Index (PCFI), Adjusted Goodness of Fit Index (AGFI) indices (> .5), Comparative of Fit Index (CFI), Incremental Fit Index (IFI) (> .9), Root Mean Square Error of Approximation (RMSEA): > .08 and Minimum Discrepancy Function by Degrees of Freedom divided (CMIN/DF) <3 good [30].

**Convergent and divergent validity.**   In this study for the estimation of convergent and divergent, we used the criterion of Fornell-Larcker [31]. The convergent and divergent validity of the BPFI was assessed using the average shared squared variance (ASV), the maximum

shared squared variance (MSV) and the average variance extracted (AVE). The convergent validity is established when AVE > 0.5 and divergent validity is established when both MSV < AVE and ASV < AVE[31].

**Reliability.** To assess the internal consistency of the Inventory, coefficients of Cronbach's alpha, Average Inter-Item Correlation (AIC) and Omega McDonald were calculated [32]. Values higher than 0.6 were considered acceptable [33]. In AIC, a good value range .2 to .4 [34]. Construct Reliability (CR), which replaces Cronbach's alpha coefficient in structural equation modeling, was then assessed and CR greater than 0.7 were considered acceptable [35].

**Normal distribution, outliers, and missing data.** The Univariate and Multivariate normal distribution of data was assessed by skewness (±3) and kurtosis (±7) [30]. The presence of any multivariate outlier was investigated by Mahalanobis d-squared ($p < .001$) and multivariate normality by the Mardia coefficient of multivariate kurtosis (< 20) [36]. Missing values calculated by Multiple Imputation technique (which were less than 3%) and were substituted with the mean scores of the corresponding items. All of the statistical procedures were calculated by SPSS- AMOS$_{25}$ and SPSS R-Menu v2.0.

**Ethical considerations.** Ethical approval for this study was provided by the Ethics Committee of Semnan University of Medical Sciences, Semnan, Iran (Approval Number: 96.524132). Necessary permissions were obtained from the developers of BPFI before its translation and psychometric evaluation. The study participants were informed about the study aim. Their agreement in responding to the BPFI items was considered to be their consent for participation in the study.

## Results

A total of 200 nursing student participants were mainly female (47.5%) and single (73%) with a mean age of 21.48±2.86 (Table 1). The mean score of BPFI score in nursing student was 48.41 ±.45. There was significant difference between female and male nursing student in resilience score (p = 0.04).

The minimum score was 23 and the maximum score was 60. After the score reversal of items 9 and 10 in the first stage of face validity, the results showed all items of the inventory to be appropriate, clear, and straightforward.

### Construct validity

In this study the KMO was .816 indicating the sample size was adequate for factor analysis and Bartlett's test of sphericity was significant ($\chi^2$ = 1283.25, df = 105; $p < .0001$), indicating that

**Table 1. Participants' socio-demographic characteristics.**

| Characteristics | | N (%) |
|---|---|---|
| Gender | Male | 93 (46.5) |
| | Female | 95 (47.5) |
| Semester | First | 55 (27.5) |
| | Third | 30 (15) |
| | Fifth | 49 (24.5) |
| | Seventh | 39 (19.5) |
| | Other | 8 (4) |
| Marital status | Single | 146 (73) |
| | Married | 43 (21.5) |

the correlation matrix was appropriate for factor analysis. The parallel analysis showed a three-factor structure for BPFI which accounted for 48.21% of the total variance (Table 2).

Next the extracted structure was assessed by CFA. The results of the Chi-square test for goodness-of-fit were first obtained as $\chi^2 = 134.420$ ($p < .0001$), and other indices were then assessed for the fit of the model. All the indices (PCFI = .760, CMIN/DF = 1.84, RMSEA = .065, IFI = .948 and CFI = .949) confirmed a good fit of the final model. For a better model, a modification was done to the final factor structure of the BPFI that there was a covary between the measurement errors of items 6 and 15 (Fig 1).

Assessing the ASV, MSV, and AVE, also indicated that the BPFI had good convergent and divergent validity. Also, internal consistency and CR of the BPFI were estimated to be desirable (Table 3).

## Discussion

This study represents the first attempt to examine the reliability and validity of a Persian inventory of resilience; the BPFI among nursing students. Previous studies conducted during the development of the BPFI support that the instrument has content, construct, and face validity, as well as reliability in measuring the health-promotion concept of resilience in adults[21].

The study was conducted in three main steps, namely that of 1). Translation and cultural adaptation, 2) data collection, and finally 3). Exploratory and confirmatory factor analysis. The result showed that the BPFI included three distinct and stable factors which explain 48.21% of the total variance in the student nursing population.

However, the original BPFI has a four-factor structure including fewer stressors, adaptable personality, supportive environment, compensating experiences. In primary EFA, three factors were extracted. In this study the factors "Adaptable personality and compensatory experiences" considered as a unit. It is probable that clients in the nonclinical group recalled their families' adaptive appraisals and compensating experiences as a unit, as well as the familial availability of these sources of competence contributing to their good adaptation[19]. Compensating experiences are defined as a family's experiences of mastery within the context of

**Table 2. Exploratory factor analysis of the Persian version of the BPFI.**

| Factor | Items | Factor loading | $h^{2\ a}$ | Eigenvalue | % of variance |
|---|---|---|---|---|---|
| supportive environment | **Q10:** I have at least one caring person in my life, (whether in your family or not). | .93 | .81 | 3.91 | 26.06 |
| | **Q9:** I have a good relationship with at least one supportive person, (whether in your family or not). | .93 | .88 | | |
| | **Q12:** I have at least one person who is interested in my life (whether in your family or not). | .80 | .68 | | |
| | **Q11:** I can trust at least one person in my life, (whether in your family or not). | .80 | .69 | | |
| Compensatory Experience | **Q15:** I have coped well with one or more major stressors in my life. | .74 | .59 | 3.36 | 22.40 |
| | **Q14:** I have control over many (but not all) events in my life. | .74 | .56 | | |
| | **Q13:** I have been able to resolve many (but not all) of my problems by my self | .61 | .30 | | |
| | **Q16:** I have been able to make "the best out of a bad situation" several times. | .56 | .33 | | |
| | **Q6:** I am a creative, resourceful, and independent person. | .49 | .33 | | |
| | **Q8:** I am competent and have high self-esteem. | .48 | .36 | | |
| | **Q5:** I am optimistic and concentrate on the positives in most situations. | .48 | .25 | | |
| Fewer Stressor | **Q4:** I have had more problems than positive experiences with work/school in the past 3 months. | .62 | .38 | 1.67 | 11.13 |
| | **Q2:** I have had more problems than positive experiences with my finances in the past 3 months | .55 | .33 | | |
| | **Q3:** I have had more problems than positive experiences with my family/friends in the past 3 months | .54 | .32 | | |
| | **Q1:** I have had more problems than positive experiences with my health status in the past 3 months. | .53 | .30 | | |

[a] Communalities

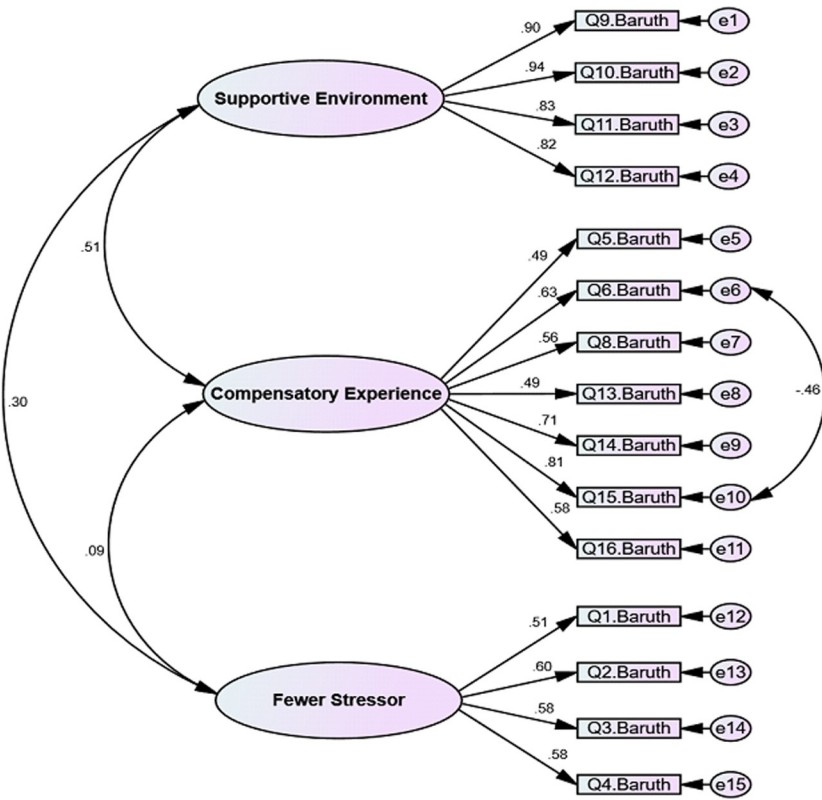

**Fig 1. Final model of the first order CFA of the BPF1.**

adversity. This mastery includes feelings of positive control over uplifting experiences while having experienced the same situations as hassles [19]. Masten (2001) posited that the influence of family mastery resources as a compensatory factor[37]. As registered nurses, all of the participants had developed compensatory and coping strategies which had enabled them to achieve their qualification and to progress in their careers[38]. However, the Adaptive resources were identified as the ability to plan and the motivation to succeed in the future, behavioral and emotional autonomy, the capacity to handle stressful situations, and access to supportive adults [39].

According to low factor loading, Items 5 and 7 were removed from the Persian version of the scale It is valuable that the differences among different cultures can result in the instability of the factor structure. Studies in the last twenty years indicated that resilience is a multidimensional concept which is affected by context, time, age, gender, cultural origin, and living conditions. In line with our findings, factor analysis in an earlier study of 410 students at a public university in the southwest of the United States revealed a four-factor structure for BPFI after

**Table 3. Convergent and divergent validity, internal consistency, and constructs reliability of BPFI.**

| Factor | α (CI 95) | AIC | Ω | CR | AVE | MSV | ASV |
|---|---|---|---|---|---|---|---|
| **Supportive Environment** | .926 (.908 to .941) | .764 | .929 | .929 | .766 | .260 | .135 |
| **Compensatory Experience** | .797 (.750 to .835) | .361 | .801 | .792 | .439 | .260 | .175 |
| **Fewer Stressor** | .657 (.572 to .728) | .323 | .658 | .658 | .325 | .091 | .051 |

CR: Construct Reliability; AVE: Average Variance Extracted; MSV: Maximum Shared Squared Variance; ASV: Average Shared Squared Variance.

the removal of item 3 [19]. Another study on the families of disabled children in Portugal also reported a four-factor structure for the scale after the removal of item 3 from the fewer stressors domain [20].

According to the final model of the BPFI, a relationship exists between the measurement error of item 6 (I am a creative, resourceful, and independent person), and 15 (I have coped well with one or more major stressors in my life). This relationship may fall, when the items are not properly understood or due to the conceptual similarity of the sentences. Coping is a protective mechanism that eliminates or modifies the situation, controls the meaning of the experience and manage the emotional consequences of the situation. Coping strategies generally seem to be more effective when an individual has more power or control over a situation [40]. Thus, this is the reason that both items have been put in the compensatory dimension of this study. Compensating experiences have been referred to as a sense of meaning and control over person-life [41] and as a person's experiences of mastery (feelings of positive control over uplifting experiences) within the context of adversity [19]. On the other hand, Compensatory mechanisms include psychological adaptations such as self-efficacy (the confidence a person to their ability to do relevant self-care activities) while the ability to cope with different situations is promoted when a person's self-efficacy is high and when person feels in control during threatening situations [42]. Children and adults who have a less negative assessment of difficult situations or who consider themselves to be in control of certain situations in their lives may be react less negatively to difficult situations and be better equipped to solve problems [43].

The factor "Social support" has been identified as a multidimensional factor that includes family, networks, community, school, employment and Consultant [19]. Social support depict medical students' resilience that experience adversity in their clinical setting [44]. In this inventory, support means a supportive environment aspect from other persons or families. A supportive learning environment promotes the resilience of nursing students, reducing their vulnerability [45, 46]. Staff support for students leads to stay student in the nursing program [44].

The factor "fewer stressors" assesses whether the person perceives more positive or negative experiences in their health status, finances, family and friends, and work or school [21]. ACHA identified stress, especially chronic stress as the number one barrier to college success for most college students [47]. Nursing students inevitably experience stress in their academic education. Resilience is a dynamic process that decreased the effects of stress and prevents the potentially disabling effects of chronic stress [44].

The results of this study also showed that the items of the BPFI have an appropriate convergent and divergent validity in the final model. The convergent validity exists when the objects of the structure are close together and detect a large variance [35]. On the other hand, it has been stated that divergent validity exists when the factors of the considered structure or the latent extracted factors are completely separate from each other. In other words, we will not have a good convergent validity when the latent factors are not well explained by the extracted items and the items are not related to each other [48]. For example, items 13, 14, and 16 refer to self-efficacy. Students who have higher academic self-efficacy beliefs use positive coping strategies to deal with academic difficulties [49].

The many studies documented that nursing students exposed to different sources of stress during their educational program [13],[18] that include long hours of study and clinical pressures [50], problem with patients or their family, problems in relationship with nurses [51], health problems [52], economic problems [53]. The BPFI has mentioned these sources of stress (Item1-4(. Also the Baruth inventory emphasized protective factors in other items, which has permitted a person to overcome adversity. Researchers have stated that nursing students use these protective factors that include support [15], optimism [17], [1], self- confidence [16], [5], Relationship [5], for coping with stress.

The coefficients of Cronbach's alpha, Omega McDonald, AIC value confirmed the internal consistency of the overall inventory. The fewer stressors dimension presented a low coefficient of internal consistency. This result might be related to the limitation of time (the last three months) and the items in this dimension (personality characteristics or environmental resources that lead to experience greater maladjustment) [20] that interact with other dimension related to personality characteristics and supportive environmental resources that help to prevent maladjustment [54].

Gender is one of the factors influencing the understanding of psychological distress [53]. The number of male nursing students is continuing to increase [55]. The resilience score in this study was different between male and female students, which was consistent with other studies [57, 56]. In these studies, the ratio between females and males were not equal but in this study, the ratio was almost the same. Also data has been collected from multiple samples (Azad university and Neyshabur university of medical science) to fairly generalize the factorial structure and construct validity of this inventory but the present study has several limitations. First, the study relied on self-reported data collected at one time point; thus, follow-up data would be valuable in assessing whether resilience results are consistent in demonstrating dispositional traits of individuals and can help health care authorities understand changes in student nursing resilience over time. Second, the current study did not measure test-retest reliability as a vital indicator of scale reliability. Therefore, future research might consider measuring other indicators of reliability, such as test-retest among undergraduate college students. Despite the study limitations, the findings from the study indicated evidence of the BPFI reliability and construct validity among undergraduate nursing students and suggest the use of the inventory to effectively assess resilience.

Further psychometric testing with other populations is warranted to enhance the BPFI's usefulness.

## Implications

In this study the first limitation was the lack of access to the sample, thus we had to run on the same sample as EFA and CFA. However we used other tests to increase the validity of the study. With fourteen items in three domains and acceptable validity and reliability, the Persian version of BPFI can be used to diagnose non-resilience among nursing students. It is shorter than other resilience tools and can be completed in less time. Thus, it can be used for resilience assessment screening purposes and also for evaluating the effects of resilience-promotion interventions among nursing students in Iran.

## Supporting information

**S1 Data.**
(SAV)

## Acknowledgments

We deeply appreciate the entire student to participate in this study.

## Author Contributions

**Conceptualization:** Tayyebeh Ali-Abadi.

**Formal analysis:** Hamid Sharif Nia.

**Writing – original draft:** Tayyebeh Ali-Abadi.

**Writing – review & editing:** Siavash Talepasand, Christopher Boyle, Hamid Sharif Nia.

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
