## [Decision Letter · Decision Letter 0]

14 Jan 2020

PONE-D-19-24033

Psychometric properties of The Baruth Protective Factors Inventory among nursing students

PLOS ONE

Dear Dr. Sharif Nia,

Thank you for submitting your manuscript to PLOS ONE. After careful consideration, we feel that it has merit but does not fully meet PLOS ONE’s publication criteria as it currently stands. Therefore, we invite you to submit a revised version of the manuscript that addresses the points raised during the review process.

We would appreciate receiving your revised manuscript by Feb 28 2020 11:59PM. To enhance the reproducibility of your results, we recommend that if applicable you deposit your laboratory protocols in protocols.io, where a protocol can be assigned its own identifier (DOI) such that it can be cited independently in the future. For instructions see: http://journals.plos.org/plosone/s/submission-guidelines#loc-laboratory-protocols

We look forward to receiving your revised manuscript.

Kind regards,

Stefan Hoefer

Academic Editor

PLOS ONE

Journal Requirements:

https://journals.sagepub.com/doi/abs/10.1177/1066480708314259?journalCode=tfja

https://www.psychiatricnursing.org/article/S0883-9417(18)30245-0/fulltext

In your revision ensure you cite all your sources (including your own works), and quote or rephrase any duplicated text outside the methods section. Further consideration is dependent on these concerns being addressed.

Additional Editor Comments:

I received two reviews for your manuscript. Based on the content of the reviews, I strongly encourage you to full address all points in a satisfying manor; in particular the statistical issues raised by reviewer 1 (e,g. CFA / EFA in the same dataset). If you are able to address these, I reinvite you to submit a substantial revision of the current manuscript. Also the organization of the manuscript, and proper proof-reeding is required.

Reviewers' comments:

Reviewer's Responses to Questions

**Comments to the Author**

1. Is the manuscript technically sound, and do the data support the conclusions?

Reviewer #1: No

Reviewer #2: Yes

2. Has the statistical analysis been performed appropriately and rigorously? 

Reviewer #1: Yes

Reviewer #2: Yes

3. Have the authors made all data underlying the findings in their manuscript fully available?

Reviewer #1: Yes

Reviewer #2: Yes

4. Is the manuscript presented in an intelligible fashion and written in standard English?

Reviewer #1: No

Reviewer #2: Yes

5. Review Comments to the Author

Reviewer #1: Feedback for the manuscript entitled “Psychometric properties of The Baruth Protective Factors Inventory among nursing students”

This study was intended to evaluate the psychometric properties of the Persian version of the Baruth Protective Factor Inventory (BPFI), including face validity, construct validity, convergent validity, divergent validity, and reliability, using exploratory factor analysis, confirmatory factor analysis, and other analyses. Overall speaking, the authors conducted many statistical analyses, but the manuscript was not written and organized well. The information in the manuscript was hard to follow. There were a lot of missing information in the method, results and discussion sections. In other words, the manuscript in the current version is immature and needs substantive revisions. A careful proofreading is also needed.

Below are some concerns, questions, and suggestions I have.

- -It seems that the authors conducted EFA and CFA analyses using the same dataset. If this is a case, it would be not appropriate to use the results from EFA and CFA with the same dataset.

- What does the software of “SPSS- AMOS” and “SPSS R-Menu v2.0.” mean?

- On page 6, how do KMO and the Bartlett’s test of sphericity check the appropriateness of the sample and the factor analysis model?

- On page 6, how to obtain the statistics for convergent and divergent validity?

- On page 8, the authors mentioned that “the results showed all items of the inventory to be appropriate, clear, and straightforward.” How did the authors make this argument? In other words, what statistical results were the authors used to make this argument?

- In addition to presenting the statistical outputs, the substantively meaningful interpretations are needed in the results.

- In Table 3 on page 10, What is AIC? What is AIC for?

- On page 11, the authors mentioned that Items 5 and 7 were removed…”. I am not sure where this cam from?

Reviewer #2: - Please give more reason/ analysis for the result, not just statistical explanation. Researcher can give the theoritical and contextual analysis. For example in line 276--287, according the concept/ theory and context/ student situation, why BPFI have an appropriate convergent and divergent validity in the final model?

- Please edit all of refferens for consistenty the style

6. PLOS authors have the option to publish the peer review history of their article (what does this mean?). If published, this will include your full peer review and any attached files.

Reviewer #1: No

Reviewer #2: Yes: Hanny Handiyani

---

## [Author Response · Author response to Decision Letter 0]

22 Jan 2020

Dr. Joerg Heber

Thank you for your thorough review and consideration of our manuscript “ Psychometric properties of The Baruth Protective Factors Inventory among nursing students" that was submitted to PLOS ONE Journal.

We believe that the comments and suggestions that were recommended by the reviewers have informed a much improved and more fully developed paper that will offer an important contribution to the field.

Given the word length requirements for clinical research briefs we are somewhat stymied by the extent of details we can provide However, and of critical importance, attention has been paid to developing a clearer rationale that addresses the significance of this work. More information has been provided to describe the measures used and data analysis conducted and this revised paper has more clarity about the participants of the study and discussion of culture.

We have highlighted the changes that were undertaken in response to your comments in the revised manuscript. A response to each of your comments is below. 

Please feel free to contact me with any questions and concerns. I look forward to hearing from you in regards to the manuscript.

Thanks for your kind attention to the manuscript.

Sincerely, Corresponding author

---

## [Decision Letter · Decision Letter 1]

8 Apr 2020

PONE-D-19-24033R1

Psychometric properties of The Baruth Protective Factors Inventory among nursing students

PLOS ONE

Dear Dr. Sharif Nia,

Thank you for submitting your manuscript to PLOS ONE. After careful consideration, we feel that it has merit but does not fully meet PLOS ONE’s publication criteria as it currently stands. Therefore, we invite you to submit a revised version of the manuscript that addresses the points raised during the review process.

While both reviewers saw merit in the manuscript, a minor of issues were raised.

We would appreciate receiving your revised manuscript by May 23 2020 11:59PM. To enhance the reproducibility of your results, we recommend that if applicable you deposit your laboratory protocols in protocols.io, where a protocol can be assigned its own identifier (DOI) such that it can be cited independently in the future. For instructions see: http://journals.plos.org/plosone/s/submission-guidelines#loc-laboratory-protocols

We look forward to receiving your revised manuscript.

Kind regards,

César Leal-Costa, Ph. D

Academic Editor

PLOS ONE

Reviewers' comments:

Reviewer's Responses to Questions

**Comments to the Author**

1. If the authors have adequately addressed your comments raised in a previous round of review and you feel that this manuscript is now acceptable for publication, you may indicate that here to bypass the “Comments to the Author” section, enter your conflict of interest statement in the “Confidential to Editor” section, and submit your "Accept" recommendation.

Reviewer #2: (No Response)

Reviewer #3: (No Response)

2. Is the manuscript technically sound, and do the data support the conclusions?

Reviewer #2: Yes

Reviewer #3: Yes

3. Has the statistical analysis been performed appropriately and rigorously? 

Reviewer #2: Yes

Reviewer #3: Yes

4. Have the authors made all data underlying the findings in their manuscript fully available?

Reviewer #2: Yes

Reviewer #3: Yes

5. Is the manuscript presented in an intelligible fashion and written in standard English?

Reviewer #2: Yes

Reviewer #3: Yes

6. Review Comments to the Author

Reviewer #2: - Please give more reason/ analysis for the result, not just statistical explanation. Researcher can give the theoritical and contextual analysis. For example in line 276--287, according the concept/ theory and context/ student situation, why BPFI have an appropriate convergent and divergent validity in the final model?

- Please edit all of refferens for consistenty the style

Reviewer #3: After analysing the study entitled "Psychometric properties of The Baruth Protective Factors Inventory among nursing students" I have observed the following issues:

In the abstract, it is concluded that "the Persian version of The Baruth Protective Factors Inventory can help nursing authorities identify non-resilient students, promote their resilience, and thereby improve their academic and clinical performance". However, there are no research findings to support this claim.

It would have been very interesting to analyze the level of stress and academic performance to see how it was associated with the validated tool.

Furthermore, although there is a table with the descriptive variables of the sample, it does not relate them to the BPFI, when it could be interesting.

Nor does it provide the descriptive data of the BPFI (mean, percentile, etc.).

The bibliography is not recent. Articles have been published in 2019 and 2020, for example, by the researcher Simonelli-Muñoz, which provide more current data related to the stress, academic performance and health of nursing students than those provided by the authors in this research.

Reference 52 should be modified.

7. PLOS authors have the option to publish the peer review history of their article (what does this mean?). If published, this will include your full peer review and any attached files.

Reviewer #2: No

Reviewer #3: No

---

## [Author Response · Author response to Decision Letter 1]

18 Apr 2020

Dear Editor-in-Chief of PLOS ONE Journal

Dr. Joerg Heber

Response to minor Revision

Thank you for your thorough review and consideration of our manuscript “ Psychometric properties of The Baruth Protective Factors Inventory among nursing students" that was submitted to PLOS ONE Journal.

We believe that the comments and suggestions that were recommended by the reviewers have informed a much improved and more fully developed paper that will offer an important contribution to the field.

Given the word length requirements for clinical research briefs, we are somewhat stymied by the extent of details we can provide, However, and of critical importance, attention has been paid to developing a clearer rationale that addresses the significance of this work. More information has been provided to describe the measures used and data analysis conducted and this revised paper has more clarity about the participants of the study and discussion of culture.

We have highlighted the changes that were undertaken in response to your comments in the revised manuscript. A response to each of your comments is below. 

Please feel free to contact me with any questions and concerns. I look forward to hearing from you regarding the manuscript.

Thanks for your kind attention to the manuscript.

Sincerely, Corresponding author 

Review Comments to the Author

Reviewer 2 Response

Please give more reason/ analysis for the result, not just statistical explanation. Researcher can give the theoretical and contextual analysis. For example in line 276--287, according to the concept/ theory and context/ student situation, why BPFI have an appropriate convergent and divergent validity in the final model?

 Convergent and divergent validity in this study was evaluated by using the approach of Fornell and Larker (1981) through ASV, MSV, AVE. p: 7

The convergent validity exists when the objects of the structure are close together and detect a large variance. On the other hand, it has been stated that divergent validity exists when the factors of the considered structure or the latent extracted factors are completely separate from each other . In other words, we will not have a good convergent validity when the latent factors are not well explained by the extracted items and the items are not related to each other. For example, items 13, 14, and 16 refer to self-efficacy. Students who have higher academic self-efficacy beliefs use positive coping strategies to deal with academic difficulties. P: 14

- Please edit all of references for consistently the style Plos One reference style download and Applied.

Reviewer3 Response

In the abstract, it is concluded that "the Persian version of The Baruth Protective Factors Inventory can help nursing authorities identify non-resilient students, promote their resilience, and thereby improve their academic and clinical performance". However, there are no research findings to support this claim. One of the research instruments that have been used to measure protective factors specifically in adults is the Baruth Protective Factors Inventory (BPFI). A significant strong point of BPFI theory is the rich theoretical basis for it. This valid and reliable inventory measure protective factors in an adult population through a review of resilience literature (resilience theories ). Assessing resilience through BPFI increases the chance of identifying concepts and variables which are important to nursing and health care practice. Therefore this variables will be clarified and important dimensions of the variables that should be measured will be determined . Therefore, BPFI is particularly fascinating in nursing students to quickly identify stress which results in providing interventions to decrease the level of stress P: 4

 Previous studies conducted during the development of the BPFI support that the instrument has content, construct, and face validity, as well as reliability in measuring the health-promotion concept of resilience in adults. p:11 

As Baruth quoted Wolff (1995) identifying protective stress causing factors would play a significant role in clinical practices. Individuals with intervene risks, have further resilience while avoiding from environmental problems. When these people got further information about the reasons and protective strategies with the help of expert ones, more resilience and stability in their life would be predicable.p:4

It would have been very interesting to analyze the level of stress and academic performance to see how it was associated with the validated tool. Certainly, It would be very interesting to analyze the correlation of stress and academic performance with the validated tool. But I didn't get permission from the designers of the tools that measure stress and academic performance.

Furthermore, although there is a table with the descriptive variables of the sample, it does not relate them to the BPFI, when it could be interesting. Nor does it provide the descriptive data of the BPFI (mean, percentile, etc.). Thank you for attention, but we only examined The demographic variables included age, gender, semester, and marriage. I write the descriptive data of the BPFI in the result. P:8

The bibliography is not recent. Articles have been published in 2019 and 2020, for example, by the researcher Simonelli-Muñoz, which provide more current data related to the stress, academic performance and health of nursing students than those provided by the authors in this research. Thank you for your attention to this valuable study. I cited it in my article. P: 3

Reference 52 should be modified.

 Plos One reference style download and Applied.

---

## [Decision Letter · Decision Letter 2]

29 Apr 2020

PONE-D-19-24033R2

Psychometric properties of The Baruth Protective Factors Inventory among nursing students

PLOS ONE

Dear Dr. Sharif Nia,

Thank you for submitting your manuscript to PLOS ONE. After careful consideration, we feel that it has merit but does not fully meet PLOS ONE’s publication criteria as it currently stands. Therefore, we invite you to submit a revised version of the manuscript that addresses the points raised during the review process.

While both reviewers saw the merit in the manuscript, a minor of issues were raised.

We would appreciate receiving your revised manuscript by Jun 13 2020 11:59PM. To enhance the reproducibility of your results, we recommend that if applicable you deposit your laboratory protocols in protocols.io, where a protocol can be assigned its own identifier (DOI) such that it can be cited independently in the future. For instructions see: http://journals.plos.org/plosone/s/submission-guidelines#loc-laboratory-protocols

We look forward to receiving your revised manuscript.

Kind regards,

César Leal-Costa, Ph. D

Academic Editor

PLOS ONE

Reviewers' comments:

Reviewer's Responses to Questions

**Comments to the Author**

1. If the authors have adequately addressed your comments raised in a previous round of review and you feel that this manuscript is now acceptable for publication, you may indicate that here to bypass the “Comments to the Author” section, enter your conflict of interest statement in the “Confidential to Editor” section, and submit your "Accept" recommendation.

Reviewer #3: All comments have been addressed

Reviewer #4: All comments have been addressed

2. Is the manuscript technically sound, and do the data support the conclusions?

Reviewer #3: (No Response)

Reviewer #4: Yes

3. Has the statistical analysis been performed appropriately and rigorously? 

Reviewer #3: (No Response)

Reviewer #4: Yes

4. Have the authors made all data underlying the findings in their manuscript fully available?

Reviewer #3: (No Response)

Reviewer #4: Yes

5. Is the manuscript presented in an intelligible fashion and written in standard English?

Reviewer #3: (No Response)

Reviewer #4: Yes

6. Review Comments to the Author

Reviewer #3: (No Response)

Reviewer #4: Title: is suitable

Introduction: it argues the need to carry out this research in a well justified manner. Perhaps it is a little long.

Material and methods: it proposes an appropriate and sufficiently robust methodology to respond to the objective of the research.

Results: expressed adequately. It is not clear if in the population represented by the sample the proportion of women is also 47%. This percentage is clearly lower than in other international contexts. It is desirable to add this aspect and discuss in the discussion section whether it has any impact

Discussion: it is appropriate, international in nature and relatively recent.

Bibliography: it is recommended to conform to the PLoS standards.

7. PLOS authors have the option to publish the peer review history of their article (what does this mean?). If published, this will include your full peer review and any attached files.

Reviewer #3: No

Reviewer #4: No

---

## [Author Response · Author response to Decision Letter 2]

11 May 2020

I revised article. Thank you for consideration. best regard

---

## [Decision Letter · Decision Letter 3]

13 May 2020

Psychometric properties of The Baruth Protective Factors Inventory among nursing students

PONE-D-19-24033R3

Dear Dr. Sharif Nia,

We are pleased to inform you that your manuscript has been judged scientifically suitable for publication and will be formally accepted for publication once it complies with all outstanding technical requirements.

With kind regards,

César Leal-Costa, Ph. D

Academic Editor

PLOS ONE

Additional Editor Comments (optional):

Reviewers' comments:

Reviewer's Responses to Questions

**Comments to the Author**

1. If the authors have adequately addressed your comments raised in a previous round of review and you feel that this manuscript is now acceptable for publication, you may indicate that here to bypass the “Comments to the Author” section, enter your conflict of interest statement in the “Confidential to Editor” section, and submit your "Accept" recommendation.

Reviewer #4: (No Response)

2. Is the manuscript technically sound, and do the data support the conclusions?

Reviewer #4: Yes

3. Has the statistical analysis been performed appropriately and rigorously? 

Reviewer #4: Yes

4. Have the authors made all data underlying the findings in their manuscript fully available?

Reviewer #4: Yes

5. Is the manuscript presented in an intelligible fashion and written in standard English?

Reviewer #4: Yes

6. Review Comments to the Author

Reviewer #4: (No Response)

7. PLOS authors have the option to publish the peer review history of their article (what does this mean?). If published, this will include your full peer review and any attached files.

Reviewer #4: No

---

## [Editor Report · Acceptance letter]

19 May 2020

PONE-D-19-24033R3 

Psychometric properties of The Baruth Protective Factors Inventory among nursing students 

Dear Dr. Sharif Nia:

I am pleased to inform you that your manuscript has been deemed suitable for publication in PLOS ONE. Congratulations! Your manuscript is now with our production department. 

With kind regards,

on behalf of

Dr. César Leal-Costa 

Academic Editor

PLOS ONE